# Functional Recovery in Parkinson’s Disease: Current State and Future Perspective

**DOI:** 10.3390/jcm9113413

**Published:** 2020-10-24

**Authors:** Manuela Violeta Bacanoiu, Radu Razvan Mititelu, Mircea Danoiu, Gabriela Olaru, Ana Maria Buga

**Affiliations:** 1Department of Physical Therapy and Sports Medicine, University of Craiova, 200207 Craiova, Romania; mirceadanoiu22@gmail.com (M.D.); olarugabriela246@yahoo.com (G.O.); 2Department of Laboratory Medicine, County Clinical Emergency Hospital of Craiova, 200642 Craiova, Romania; 3Department of Biochemistry, University of Medicine and Pharmacy of Craiova, 200349 Craiova, Romania; razvanmititelu@rocketmail.com or

**Keywords:** Parkinson’s disease, rehabilitation, physical exercises

## Abstract

Parkinson’s disease (PD) is one of the most frequent neurodegenerative disorders, affecting not only the motor function but also limiting the autonomy of affected people. In the last decade, the physical exercises of different intensities carried out by kinetic therapeutic activities, by robotic technologies or with the participation of sensory cues, have become increasingly appreciated in the management of Parkinson’s disease impairments. The aim of this paper was to evaluate the impact of physical exercises with and without physical devices on the motor and cognitive variables of PD patients. In order to achieve our objectives, we performed a systematic review of available original articles based on the impact of kinetic therapeutic activity. Through the search strategy, we selected original papers that were laboriously processed using characteristics related to physical therapy, or the tools used in physiological and psychological rehabilitation strategies for PD patients. In this study, we presented the most current intervention techniques in the rehabilitation programs of patients with Parkinson’s disease, namely the use of assisted devices, virtual imagery or the performing of physical therapies that have the capacity to improve walking deficits, tremor and bradykinesia, to reduce freezing episodes of gait and postural instability, or to improve motor and cognitive functions.

Tel.: +40-0351-443-500 (A.M.B.)

## 1. Introduction

Parkinson’s disease (PD) is the most common disease in terms of gait disorders, bradykinesia, tremor, postural instability and balance [1,2]. Patients with PD can develop non-motor symptoms including emotional functions, sleep disturbances, and psychiatric symptoms such as depression and anxiety [3]. Non-motor symptoms may be more negative than motor symptoms as they affect the patient’s life [4]. Levodopa therapy has recognized benefits but considerable side effects [5,6].

PD symptoms affect various aspects of life, including the ability to transfer and walk, leading to disability [7]. Improving the quality of life of patients with PD, the role of physiotherapy is to maximize functional capacity, independence, safety and well-being, thus reducing secondary complications and also addressing the fear of falling and maintaining physical activities [8,9].

As a neuropathological mechanism, PD involves the depletion of pigmented dopaminergic neurons of the substantia nigra pars compacta from the black matter of the brain stem with the accumulation of pathological proteins, called Lewy bodies, until the alteration of cortical–thalamo–circuits.

Thus, motor impairment or parkinsonism becomes the fundamental component of the evolution of the disease, both in its premature phase and in the advanced phase. The four cardinal motor points in PD (tremor, bradykinesia, rigidity and balance disorders), apart from pharmacological therapy, can be traced but also improved through physical rehabilitation programs.

Tremor can be divided into two different patterns: “rest tremor”, which occurs as subjects relax their muscles, and “action tremor”, which occurs while subjects perform voluntary muscle movements. There are two types of action tremors: postural and kinetic tremor. Postural tremor occurs when a person holds his hand or hands or feet against gravity and kinetic tremor occurs when a person is engaged in various daily activities that involve voluntary muscle movements. There are several causes of tremor. Idiopathic Parkinson’s disease (IPD) is the most common cause of tremor of rest, although it can also cause tremor of action [10].

Parkinson’s disease (PD) is associated with gait disorders, instability postural and freezing of gait [11,12]. These gait deficiencies are associated with significant disabilities because they lead to reduced mobility, falls and injuries caused by falling [13,14]. Freezing of gait (FOG) in PD is a short episode of absence of forward progression of the legs, despite the intention to walk. Patients have the impression that their feet are glued to the ground [11] and they lose control over their gait. FOG is an episode which lasts from a few seconds to up to 1–2 min [11,15] and occurs when movement is initiated near narrow spaces or on return [16,17].

Increased muscle tone affects both the flexor and extension muscles, is evidenced by passive movements (limbs, throat, trunk) and is emphasized by active voluntary motion. With the PD progression, other impairments occur, such as postural instability and gait dysfunction [18,19]. Due to these deficiencies, there is a high tendency to fall but also a reduced ability to walk, which determines the reduction in motor function [20]. All the physical interventions contributed to enhancing the quality of life (QoL) in patients with PD and to restore the damaged functions [20,21,22,23,24].

Rehabilitation through physiotherapy, in recent years, was focused on decreasing gait impairments and improving cognitive parameters [25].

Physical activity provides an essential contribution to increasing walking parameters, balance and muscle strength for patients with PD. In this light, specialists in the field consider this intervention to be essential, due to its contribution to motor function recovery [26].

## 2. Materials and Methods

### 2.1. Search Criteria

In order to recognize the relevant publications, we searched PubMed and Cohrane electronic databases for all papers, using as keywords: “Parkinson’s disease”, “rehabilitation” and “physical exercises”. The selected reviews, meta-analyses, clinical trials, pilot study, original papers were sorted accordingly with PRISMA flow diagram.

In this study, original articles relevant to the modern and technological methods used in the rehabilitation therapy in PD were considered. The most representative full-text studies published in the last 10 years were scanned. All these studies used assistive equipment for the physical training of patients with mild to moderate PD disease. Only the materials in English were taken into consideration.

### 2.2. Selection Strategy

Some guidelines have been used to make this synthesis: (1) the study groups with idiopathic Parkinson’s disease were selected in adults belonging to the different severity scales of the disease; (2) the participants of both sexes were randomized and all completed the physical intervention; (3) the physical training used both passive exercises and active exercises (mobilization of the joints, voluntary or endurance exercises, or even daily activities); (4) all patients included in the studies had conventional anti-PD medication; (5) the training times were from daily domestic activities or occupational therapy to physical exercises with 5–10 min of warm-up and 40–50 min of physical training with different frequencies and different development periods; (6) in the vast majority of cases, physical therapy was accompanied by participation via assistive devices that imposed a certain frequency and a certain rhythm to the range of motion; (7) the motor benefits and motion rehabilitation after physical training, were mainly considered, but the non-motor variables were also included in the final results; (8) participants in the randomized trials were separated into experimental and control groups. Criteria for exclusion from the study were: other neuromuscular disorders or chronic cardiovascular disease, diabetes mellitus, various surgeries, and patients with framework support or those who needed permanent assistance.

### 2.3. Dynamic Extraction Data

We performed a full-text keywords searching across the main databases in the last 10 years, and subsequently selected relevant publications by applying the selection criteria. A total of 1264 papers were identified from PubMed and Cochrane databases. All these papers were addressed to PD rehabilitation using physical exercises. After the elimination of duplicate studies, the number of papers was diminished to 918, which were scanned, and 605 of these were excluded. A total of 313 original full-text articles were ultimately considered eligible. Animal models, preliminary data, case reports and writings that did not use physical exercises were excluded. Finally, 39 representative papers were included for the qualitative analysis for the purpose of our study.

### 2.4. Pattern Quality Assessment

The extracted and verified data included the following variables: (1) reference and data publication; (2) framing the disease evaluated through motor and non-motor scales; (3) using assistive devices in physical therapy; (4) features of the intervention (pattern of physical exercise, frequency, intervention time of sessions, motor and non-motor variable measures); (5) results, discussions and conclusions observed in each paper.

## 3. Results

The organizational chart that shows the dynamics of the research is presented in Figure 1 PRISMA diagram.

### 3.1. Strategies to Improve Functional Outcomes in PD Patients

PD is the second most neurodegenerative disease, which affects, on average, 3 in 100 older people (over 65 years) [27,28]. Some studies estimate that there will be a doubling in number by 2030 [27,29]. Due to a decreased number of dopaminergic neurons from substantia nigra and the formation of Lewy bodies, the motor (e.g., resting tremor, extrapyramidal impairments, instability postural, gait disorders) and non-motor functions (e.g., neuropsychiatric disorders, cognitive decline, anxiety, fatigue and depression) can be severely affected [30,31].

The tremor appears at the beginning of the disease, which manifests itself in a state of rest and diminishes during active movements. The tremor can be accentuated by stress and rigidity, which manifest at the level of the flexor and extensor muscles. These factors increase the resistance to the passive movements. Subsequently, bradykinesia is installed, which manifests itself by diminishing the movements and decreasing their amplitude, but also by the postural instability that is associated with balance and coordination disturbance. The postural tone is modified in the sense of a flexion tendency. The PD patients are rigid, with an abnormal posture, which is maintained during the walking activities that are performed with small steps, the festination (gait acceleration) is also present and the patients stops upon impact with an obstacle.

The appearance of disability in PD is the cumulative effect of gait disorders, posture and balance and, as the disease progresses, the steps become even smaller until the gait is blocked by freezing phenomena. The anterior flexion of the trunk also accentuates the danger of falling by moving prior to the center of gravity, and posture reflexes are seriously affected.

Walking disorders primarily affect the quality of life of patients with PD, especially in the course of the disease. Therefore, in addition to the drug therapy imposed in PD, the neurorehabilitation programs promoting the improvement in walking disability have gained ground. The most important symptoms such as weakness and fatigue affect the functional ability to walk, and therefore diminish the quality of life [32,33,34].

The major concern in PD is to better quantify the motor symptoms and impaired cognitive function and to find new ways to improve the functional outcome. In this light, it is important to use the most relevant tools in order to proper estimate the level of functional and cognitive deficits and to monitor the functional recovery.

Therefore, some instruments are used to quantify motor and non-motor disorders in PD patients in a complex mode. Other, more specific subscales are frequently used to quantify more specific tasks in PD people. The available tools are summarized in Figure 2.

Unified Parkinson’s Disease Rating Scale (UPDRS), established in 1980, was the main tool used to show the level of disability, using four classification scales of signs and symptoms, from lowest to highest disability status [35].

An alternative to the UPDRS scale is the Hoehn and Yahr scale that quantifies the disease progression into five levels [36]. The Hoehn and Yahr scale was used for the first time in 1967 and modified with additional intermediary steps [37].

Functional Independence Measure (FIM), with motor and non-motor tasks is an available complex tool used to quantify the degree of functional independence during the rehabilitation process in stroke, traumatic brain injury (TBI) or PD patients [38].

The motor performance can be assessed using focused tools. Some scales are available for gait, balance and posture evaluation during the rehabilitation process. Berg Balance Scale (BBS) [39] with a score range from 0 to 56 (excellent), which quantifies balance tasks, is the gold standard tool used to evaluate the balance and posture in a static and dynamic mode. The BBS scale is a reliable tool that can be applied in order to measure the functional outcome in a range of diseases, including PD. Alternatives to BBS include: (i) Tinnet’s Performance-Oriented Mobility Assessment (POMA), where the better score is 28, which evaluates static and dynamic tasks [40]; (ii) the sit-to-stand test (STST), which counts how many times the patient sits and gets up in 30 s [41]; (iii) 10, 6, 5-Meter Walk Test (10, 6, 5 MKT), which evaluates walking for 10 m with the highest speed [42]; (iv) Activities-specific Balance Confidence (ABC), which quantifies posture and trend of falling [43]; (v) Range of Motion (ROM) [44]; (vi) Dynamic Gait Index (DGI) [45]; (vii) Postural Instability and Gait Disability (PIGD), which evaluates gait and posture disorders [46]; (viii) Balance Evaluation System Test (BESTest), which assesses the performance balance of the patient [47]; (ix) Frontal Assessment Battery (FAB), which evaluates motor functions [48]; (x) Human Activity Profile (HAP), where a score higher than 74 indicates an active patient [49,50]; (xi) Kellogg International Work Group on the Prevention of Falls in the Elderly [50,51]; (xii) Cumulative Illness Rating Scale/SPDDS (CIRS), which quantifies faller status [52], (xiii) Pastor Test (Shoulder Tug), which evaluates the stance of a patient at different obstacles, (xiv) Physical Performance Test (PPT), which evaluates motor performance [53].

However, only some of these measurement tools are specifically addressed to elderly people with PD: (i) The Timed Up-and-Go (TUG), which is assessed using a stopwatch, and measures the ability to walk for 3 m then turn back and sitting on the chair again [54]; (ii) Physical Activity Scale for Elderly (PASE) [55]; (iii) Tinetti Gait and Balance Instrument, which shows an elder’s risk of falling, where a lower score means severe impairments.

PD is a multifaceted disorder that affects not only motor function, but also cognitive function and behavior. The available instruments used to quantify non-motor or cognitive disturbances are focused on different tasks. For cognition quantification, some valid scales are used: (i) Montreal Cognitive Assessment (MoCA), which performs a screening for cognitive impairments [50,56]; (ii) Priscus-Physical Activity Questionnaire (PRISCUS-PAQ), which assesseses how patients with PD spend their daily activities and rest times [57,58]; (iii) MMini Mental Scale Examination (MMSE), which evaluates cognitive function disorders [59,60,61]; (iv) Patient Global Impression of Change (PGI-C) [46]; (v) Parkinson neuropsychometric dementia assessment (PANDA); (vi) Hamilton Rating Scale for Depression (HAM-D); (vii) Beck Depression Inventory-II (BDI-II), with a score from 0 to 63, which investigates the level of depression, where a higher score means severe depression [62,63]; (viii) Hospital Anxiety and Depression Scale (HADS), which quantifies depression and anxiety in patients with PD [64].

In order to evaluate the impact of diseases on quality of life, some questionnaires are validated for PD patients: (i) Parkinson’ Disease Questionnaire with 39 items (PDQ-39), which investigates the quality of life [46,57,65]; (ii) Fatigue Severity Scale (FSS), which estimates fatigue in 5 min through severity disease, (iii) Priscus-Physical Activity Questionnaire (PRISCUS-PAQ), which assesses how patients with PD spend their daily activities and rest times [57,58]; (iv) PD Sleep Scale (PDSS), with 15 items of a visual scale, which refers to quality of sleep for PD patients [61,66]; (v) Schwab and England Activities of Daily Living (SD-ADL), which assesses daily life and functionality [44,46,67]; (vi) EuroQol Quality of Life/Quality of Life Visual Analog Scale (QOL/QOL VAS) [62] (vii) Patient Global Impression of Change (PGI-C) [46]; (viii) Physical Activity Questionnaire (LAPAQ), which summarizes questions about daily activities and two sports activities [44].

The cognitive and quality of life outcome of aged PD patients should be carefully assessed using tools that are validated on aged PD patients. Here, we only have some available tools that are validated on aged people: (i) MMSE and (ii) Geriatric Depression Scale (GDS), which reflects depression [65,68].

While PD is a neurodegenerative pathology with a progressive evolution and gait disorders, altered postural reflexes as well as limited joint mobility and impaired balance are associated with disorder cognitive functions, which justifies the major concerns for ameliorating these deficiencies and supporting better living conditions. Therefore, the involvement of personalized rehabilitation therapy using specific complementary tools is proven to improve the evolution of the disease and the quality of life of the patients with PD.

#### 3.1.1. Impact of Non-Assisted Recovery Strategies with Instruments of Quantify the Motor and Cognition Functions

The main characteristics of physical therapy such as duration, frequency and exercise in rehabilitation of patients with PD, are presented in Table 1.

In the six articles [57,61,65,70,71,72] the authors discuss activities of daily living, such as physical exercises, motors and cognitive results obtained after the application of the therapy.

For instance, activities of daily living (ADLs) with different degrees of effort (physical rets, moderate or stronger physical exercises), such as household activities, gardening, walking outside, cycling, gymnastic or tidying, or passive activities such as reading, watching and sewing, which were performed for 2 h for 7 days, with a twice/week frequency, significantly improved motor functions and subjective conditions for patients with PD [59]. Another intensive rehabilitation protocol, referring to stretching exercises, relaxation, gait and balance training for 4 weeks with three sessions/day for 5 days/week, resulted in important improvements in gait, balance, range of motion and important improvements if the quality of sleep [61]. After 2 or 4 weeks of exercise for daily activities, significant improvements in range of motion, gait, balance were observed. This improvement decrease the fear of falling, depression and anxiety [72]. Also, studies proved that the long term exercise can promote a delay in motor function deficiencies progression, or can promote reconversion of the 4th stage to the 3rd stage (UPDRS criteria). Other studies reported an improvement in the FIM and BBS scales and in the quality of life, after long-term guided physical exercise [70].

Performing daily physical activities without specifying their time or frequency has brought benefits in terms of the rehabilitation of motor functions, especially for the lower limbs, related to bradykinesia, and improved verbal expression through speech therapy [65].

Under the conditions of performing physical therapy through regular aerobic exercises for 12 weeks and 4 weeks, respectively, improvements in balance and motor functions were observed by decreasing the level of serum homocysteine [67] and decreasing neuroinflammation by involving T cells [69]. Improving walking parameters, range of motion, better mobilization and reducing FOG episodes was achieved by promoting walking training such as 20 m long and 3.5 m wide postural and stretching exercises, performed for 10 weeks in 16 sessions, which took place twice a week with one hour of exercise. Under these conditions, we observed walking speed improvements, which facilitated sitting and standing transition and increased ROM and FOG. For evaluation of cognitive functions, the MMSE and FOG Questionnaire were used, which confirmed the beneficial results when the scores were assessed after performing the training program. The motor function measurement tools (e.g., UPDRS III, Hoehn and Yahr, ROM, SPPB) showed an improvement of motor function after minimum 2 days per week of physical training in group. This improvement in motor scales was further sustained by improving in non-motor scale (MMSE and FOG) that [44]. Through gait self-speed training for 12 m and the Time up and Go (TUG) protocol, which involves walking for 3 m, turning, walking back and then sitting down, benefits were obtained in terms of improving gait speed, especially for the lower limbs. The evaluation of non-motor functions was performed through MMSE and SD-ADL [72]. Extended physical therapy for the whole body (cervical area, trunk, upper and lower limbs) through speech, face, breathing and postural re-education exercises, dance exercises, stretching therapy, balance and gait training, improved the range of motion and postural disability, including gait and balance. Measurement of clinical criteria was performed by: UPDRS III, Hoehn and Yahr, PPT and Pastor Test (Shoulder Tug) [53]. Another physical exercise for joint mobilization, decreasing rigidity, postural exercises, breathing therapy, ergotherapy, was applied, they resulted in an increase in the range of motion and quality of life. The following clinical assessment methods for motor symptoms were used: gait score (GS), functional independence measurement (FIM) and Tinetti Gait and Balance Instruments [52]. Because this neurodegenerative disorder is associated with a reduction in the frequency and amplitude of movements, called bradykinesia, which also affects the cephalic area, swallowing and aspiration disorders were mentioned in this regard. Parkinson’s disease is a neuromotor disorder that affects the striated muscles of the oral cavity, thus affecting the organs that are responsible for swallowing, breathing and nutrition, causing dysphagia [74]. For this reason, the intervention of rehabilitation through physical therapy based on a complex swallowing protocol is an important objective in PD diseases management. These protocol can improves swallowing and respiration and delays the progress of dysphagia. Accordingly, the application of an orolingual exercise (OLE) program three times: two exercises swallowing saliva, two exercises of tongue protruding out and two exercises of tongue rolling back, brought significant benefits in terms of the coordination of swallowing and breathing [73].

#### 3.1.2. Impact of Assisted Recovery Strategies with Instruments for Quantifying the Motor and Cognition Functions

In our study, we will further discuss the intervention of tools used in physical rehabilitation, which have significantly contributed to the amelioration and delay of the onset of motor disorders and cognitive symptoms associated with PD.

The main interventions in PD patients and their characteristics, the clinical instruments used and the associated motor and non-motor benefits are presented in Table 2.

Motor deficits in PD are responsible for postural instability and freezing-of-gait episodes. The temporal organization stride registration was performed using a one-dimensional accelerometer (Vitaport 3) through physical activity, which means normal walking training for 42 m around an oval inner track. The session lasted 10 min, after which the motor results such as gait variability, especially speed and postural instability were interpreted using the following tools: UPDRS, ABC and BESTest scales. The use of an inertia–motion device is an interesting perspective which helps to quantify the walking cycles and to appreciate the balance pattern [75].

Martelli and collegues showed that using an active tethered pelvic tool (A-TPAD) in a 30 min session of walking, with perturbed training with different speeds on treadmill monitoring, can significantly improve ability and stability, gait and balance. Active physical intervention improved postural disturbances, quantifying reactive balance and decreasing risk of falling, due to the use of the following motor instruments: UPDRS, Hoehn and Yahr, TUG and 5-MKT [76].

PD is associated with slowing down movements and accentuating the postural tone, which imposes various degrees of rigidity. Therefore, the major objective of rehabilitation in PD patients is focused on identifying some physical exercises and the optimum time frame and frequency that can allow as to obtain a significant improvement of functional outcome. By using rebound therapy exercises and weight-supported exercises through Biodex Isokinetic device (trampoline) for 8 weeks with 20–45 min per session and three sessions per week, motor benefits were obtained: enhancing contraction force for lower extremity muscles, improving intervals of motion, proprioception and, as a non-motor result, increasing quality of life [55].

For the six original studied articles, we recorded data on the use of physical activity of patients with PD, and the cycle ergometer with or without the participation of sensory cues. The exercises per session with stationary bicycle lasted from 9 to 40 min and, as a time interval, from one day to 8 weeks. Physical training was represented by endurance exercises through high-cadence cycling [62], with major benefits to non-motor variables, such as improving emotional cognition and quality of life, with the participation of the following non-motor instruments: MoCA, BDI-II, QOL/QOL VAS. Using a pedaling rate with low-intensity cycling for 8 weeks, for 15–40 min per session with 2 or 3 sessions per week, the performance of the gait and motor functions was improved. However, the mechanism by which this pattern of exercise act to improve the motor function is still unknown. Also, these studies were performed using a small number of patients and should be further validated in a large cohort [46].

Sustained endurance training (leg press, ankle plantar flexion, chest press) for 12 weeks with 50 min per session and two sessions per day with resistance therapy and the same load, or five sessions per day of endurance exercises with progressive load from low-intensity, high volume to high-intensity and decreased volume, postural instability and life conditions were improved [47].

With a pedaling rate for 8 weeks for 40 min per session, positive additional connections between thalamus and cortical motor areas were noted, with efficient results for balance and postural reflexes [77].

Under the conditions of associating the stationary cycle with acoustic (metronome) and visual cues (virtual reality), rehabilitation of motor functions and a decreasing rate of falling were noted [78,79].

The use of the dynamometer as an auxiliary means in strength training for 4 weeks, for 60 min per session, led to improvements in pattern gait, balance and postural stability [91].

Gait analysis was completed using Gaitre System (pressure sensitive carpet) for normal walking training for 22 m, of which 4.88 m were on a pressure-sensitive carpet and 17.12 m were spent walking self-speed for 15 min per session. Repetitive step training, postural functions, gait skills and spatiotemporal gait characteristics were improved [80,81].

Robotic gait training improved the kinematic parameters of GVSs (pelvic tilt, rotation, hip flexion, abduction, rotation, knee flexion, ankle dorsi-plantar flexion, foot regression) band length, velocity and cadence steps, using the assistive device training, GE-O System, for 4 weeks with 45 min per session and 20 sessions per week [82]. As Parkinson’s disease progresses, the difficulties in initiating movement and maintaining gait are increasingly obvious. These disturbances were characterized by the presence of freezing of gait episodes.

The FOG assessment was performed using the FOG questionnaire, patient statements or by analysing their gait. Soltaninejad S. et al. [83] used KIN-FOG, which is an automatic system which simulates Freezing of Gait in order to anticipate the occurrence of these episodes.

Physical therapy proposed two walking patterns: simple walking (SW) for 2.64 m carried out in a single session, and walking and turning (WWT) in two sessions (back and forth). This tool has proven an accuracy of over 90 % for anticipation of FOG episodes [83].

In this regard, a physical activity of walking for 4 m, with normal speed walking, normal and rapid speed walking with short steps, and turning gait in both directions (360°) with a secondary task (rubber ball held in the most affected arm), was performed. The motor instruments used were UPDRS, PIGD and FAB, and the non-motor used tool was the FOG questionnaire. The motor outcomes demonstrated that turning gait required additional time for the secondary task during walking, delaying the time for the installation of FOG [86].

Gait self-speed training with robot-assisted Lokomat-RAGT for 4 weeks, with 30 min per session and five sessions per week, for PD patients, demonstrated improvements in gait, especially speed, and enhancements of cognitive parameters (MMSE > 24) [59].

For investigating balance, tremor, gait and postural instability, a multidirectional perturbation platform was used, with one disturbance protocol in twelve directions. In this regard, the correlation between antagonist muscles of lower limb with reactive stance was highlighted [45].

By means of an implanted DBS neurostimulator, local potentials were collected from the subthalamic nuclei (STN) during physical training, which comprised sitting and standing or resting, and 125 m of walking at self-speed (back and forth). Biopotential recordings showed beta band oscillations, a situation correlated with rigid symptoms [91,92,93]. At the same time, stimulation of the STN while walking brought improvements in gait-cycle parameters such as speed, cadence and stride length [84].

Turning monitoring (turn mean, velocity, amplitude, duration, cadence) during activities of daily living (ADL) inside and outside, for 10 h per day and 7 consecutive days, was demonstrated by using opal inertial sensors at each foot and the belt [85].

Repetitive stepping in four directions (forward, back, left and right) for 4 weeks, for 15 min per session and three sessions per week, using Smart EquiTest Balance Master, improved pattern gait, balance reactive and postural stability, and decreased fall trends [79].

ADL were monitored for 6 months, with 20 min per session and three sessions per week, using Step Watch 3 Activity Monitor on older adults with Parkinson’s disease. Motor outcomes showed an improvement in gait balance, resistance and flexibility, and a decrease in walk disorders while the non-motor outcomes proved an enhancement of the quality of life and expressed fear of falling [87].

Dysgraphia features of writing that are due to bradykinesia, especially micrography, are known in Parkinson’s disease. Handwriting training carried out over 6 weeks, with 30 min per session, and five sessions per week, showed an improvement in writing size and fluency but unchanging writing velocity [60].

In physical rehabilitation of PD patients, a treadmill BWSTT/Bertec was used, either associated or not associated with other robotic tools. The duration and frequency of physical exercise was different, beginning with one day for 8 weeks, with 6 min to 45 min per session, and one to 20 sessions per week. Regarding the weight-bearing supported exercises on the treadmill training group, an improvement was obtained in the interval of motion for lower extremity muscles, proprioceptive reflexes and environmentally friendly life. The monitoring was performed through motor instruments such as UPDRS and Hoehn and Yahr and a cognition tool of the type Parkinson’s disease questionnaire [55]. Through self-speed walking or increasing-speed- or overground-walking training using a treadmill, improvements were obtained in gait cycle, reactive balance, postural stability, gait pattern, decreasing the risk of falling and enhancing cognitive parameters [59,76,79,82,88]. Neurocognitive assessment battery (WEB Neuro) in endurance exercises, for one week, with 40 min per session and three sessions per week, obtained important benefits in the investigation of neurocognitive functions and emotional recognition [62].

In sight of estimating the tremor severity and steps recorded during overground walking, movement sensors were used in daily living activities for 120 min per session and two days per week [88,89].

Through the intervention of video games such as Nintendo Wii and Wii Fit games, the tool named WII FIT balance has managed to bring an improvement in gait performance and monitoring of the reactive balance [90], mediated by a 6-week exercise program, for 20 min per session and two sessions per day.

#### 3.1.3. The Future of Assisted Recovery Strategies Using Tele-Medicine

Tele-rehabilitation is a promising tool that can be extensively used in the future in order to achieve a good functional recovery, not only in PD patients but also in other diseases like stroke, multiple sclerosis, diabetes or cancer diseases. Despite the recent advances in the tele-neurology field, there is still a gap in the knowledge regarding the aging patient’s perception and user compliance between the elderly people. The previous studies were focused on stroke diseases [94], or were performed using young subjects [95], but little is known about other neurological diseases such as PD or about the efficiency of this strategy on the aging population. One study showed that tele-medicine can improve the neurological healthcare, but this study was performed using middle-aged people (age mean of 37.8 years) [95]. Some recent studies focused on the telemedicine benefits in PD patients. One study showed that we need to rehabilitate the voice of PD patients due to various articular dysfunctions in speech that impair interpersonal communication and verbal intelligibility, and this requires the intervention of specific programs such as Lee Silverman Voice Treatment (LSVT) or LSVT Loud that involve speech exercises mediated by technological competence, telerehabilitation, or questionnaires that can be performed without physical participation from the patients with PD [96].

This is very important because this approach does not refer to sex, gender, age and disease stage and does not influence the distances at which patients are located, as well as benefiting from low costs given that, in PD, there are a number of motor difficulties that seriously affect motor skills and the ability to move these patients [97,98].

Through Clinic Head (Human Empowerment Aging DNA Disability) or Home Head programs, significant improvements were obtained in motor and non-motor deficiencies such as improved motor skills, obvious progress in balance and patient gait and in the mobilization of the upper and lower limbs, but also in improving cognitive functions, memory, positive affect and mental health, and implicitly in improving the quality of life of patients with PD, regardless of the age, sex or evolutionary stage of the disease in patients with PD [99].

However, there are limitations to the application of tele-rehabilitation due to the association of the disease with various deficiencies that prevent its application, such as: poor manual dexterity, difficulties in auditory and visual indicators, deficient positions or education level [96]. This prospect is considerable in terms of multidimensional functional recovery due to the fact that patients can develop effective remote programs in ameliorating motor and non-motor symptoms with considerable improvement in quality of life and lower costs to society through telerehabilitation in PD patients [99].

## 4. Discussions

The purpose of our study was to synthesize interventions of physical rehabilitation protocols, either classical, assisted or through the participation of sensory cues, as well as the motor and mental results observed after their application. Physical therapy applied in non-assisted strategies has used physical recovery schemes, ranging from activities of daily living to endurance exercises. Also, multimodal exercises, real walking protocols, TUG protocol or static physical therapy (e.g., joint mobilization, stretching, balance and postural training) were predominantly used in these studies. Only in two interventions focused on cephalic extremity, the impact of orolingual exercise program [73] and swallowing management protocol [74] were analyzed. This program was aimed to optimize swallowing functions, improving dysphagia and leading to better timing between breathing and swallowing. Through ADL, mimic, face and speech exercises were performed [52]. At the level of the thorax and upper limbs, exercises which stimulated rhythmic rotational movements, stretching of the flexor muscles, breathing exercises and occupational therapy were applied. For pelvic extremity and lower limbs gait training was initiated with self-speed, increasing speed and turning, an intervention which improved the range of motion, reactive balance, motor functions, sitting and standing transitions. These benefits were highlighted through the improvement in motor outcomes such as BBS, FIM, ROM and FOG.

In another twenty-five studies, which were accessed via robotic assistance and virtual imagery, we analysed the impact on motor and cognitive functions. From the study of original articles which used a treadmill associated with different tools, we extracted positive interventions regarding motor and emotional rehabilitation. We highlighted important results such as improving the gait pattern, contraction force, especially in lower extremity, enhancing proprioception, balance, postural stability and decreasing the risk of falling, which were obtained through weight-supported exercises, self-speed walking or increasing speeds. Through analysing original papers which discussed using a cycle ergometer with or without auditory and visual cues associates, we concluded the importance of this assistive device in motor and cognitive rehabilitation. Looking at substrate endurance exercises which were carried out at high or low intensity, with the participation of virtual imagery, we noted an improvement in gait performance, alleviation of bradykinesia, as well as optimization of pulse conduction at the thalamic level to cortical motor areas. Furthermore, we observed an enhancement of neurocognitive and emotional functions as well as an improvement in the quality of life. Participation imagery training or virtual games have been recommended by some authors, as well as additional therapy focused on motor learning and control strategies without involving physical exercises, but these benefits should be further validated on ageing patients cohort [100]. We observed an improvement in gait kinematic, gait cycle, modulation of postural disturbances, and a reduction in the risk of falling or an augmentation of cognitive parameters, which were mediated through robotic gait training with the participation of assistive tools (GE-O system, GAITRite system A-TPAD, Lokomat-RAGT). In a few studies, different types of sensor were used, such as motion sensors, opal inertial sensors or neurostimulator DBS, placed on the limbs, belt or head, with stimulatory actions which harvested local potential from those areas. We noticed real benefits in improving gait speed, cadence, stride length, gait balance, monitoring the turning and, very importantly, in estimating tremor severity. We observed that the other devices used were favorable in improving reactive balance (multidimensional perturbation platform, Smart EquiTest Balance Master, WII Fit Balance board), postural instability, gait performance and decreasing fall trends, through stepping or walking training in four directions (forward, backward, left and right).

On the other hand, physical activity, which was associated with a second task (ball with sensor pressure held in the hand) needs additional time for workout, which delays the time for FOG installation [86]. The research work focused on studying the impact of writing training on the motor functions of the upper limb reported an improvement in writing size and fluency, but unchanging writing velocity [87].

By processing original articles, we concluded that there was a beneficial effect in the motor field and mental area from both conventional and additional physical participation, through assisting tools. Also, complementary physical participation through virtual images and even auditory stimuli can be a valuable instrument in the future that can be used by PD patients from the comfort of his home. 

However, we found that there are limitations to each of these perspectives, as some of the study data are heterogeneous, and the parameters characteristic for the applied physical exercises are not uniform.

Today, in this new pandemic situation, the importance of the tele-health network should be reconsidered. This network allows us to have the best expertise of specialists in the field, not only for diagnostic purposes but also for the rehabilitation. On the other hand, these services are currently limited in time and are difficult to access due to insurance limitations. In this light, clinicians spent a lot of precious time guiding patients in accessing these services from health authorities. Even in this case, it is well known that a significant portion of neurological patients, including those with PD, cannot access the rehabilitation places due to significant functional deficits in the advanced stage of the disease. However, for low- or middle-income countries across Europe, another significant barrier is represented by the financial support of PD patients from local health politics. All countries across the EU should combine their expertise, infrastructure and training in order to achieve the best personalized strategies for neurorehabilitation, not only in PD but also in other neurological diseases. The tele-heath network across the EU countries can offer a best alternative to the conventional local rehabilitation management, but a disadvantage can be limited access to all categories of patients independent of level of education, instruction or language barrier.

All these interventions were restricted by financial possibilities, educational level and cooperation in performing this modern therapies. Therefore, some perspectives are outlined in this regard, which will be addressed in the next section.

## 5. Conclusions

Physical therapy in all its forms has clear beneficial effects in the motor and cognition rehabilitation of Parkinson’ disease.

It is worth mentioning that the types of approached therapies led to significant improvements in gait, including improvements in speed, cadence, velocity, strength step, and turning training, balance, postural reflexes, range of motion, breathing, swallowing, mimic, writing and speech therapy, as well as a delay in freezing of gait and cognitive functions.

We would recommend the continuation of the classic physical therapy, but with a decrease in the duration and increase in the frequency of the exercises, especially ADL, which allows the modulation of household activities.

When using assistive tools, which have faster and more efficient results, but are more expensive, the recommendation is to continue this variant and to find opportunities for continuous therapy at home and not in rehabilitation centers.

A future perspective is the most active use of virtual imagery and auditory sensors associated with physical activity, at home, through games, or visual programs used on PC, smartphone or TV.

The aim is to compile as efficient an algorithm as possible, in terms of motor rehabilitation and mental functions in PD.

In the future, we aim to compile an algorithm which standardizes the most efficient methods, through motors and cognitive function strategies, and possibly the combination of the two, leading to the best solutions to raise the quality of life in patients with PD. There is also an urgent need to build a European tele-rehabilitation network that can be easily accessed by PD patients all over the world from the comfort of their homes, which should offer a multidisciplinary strategy for complex diseases such as PD.

## Figures and Tables

**Figure 1 jcm-09-03413-f001:**
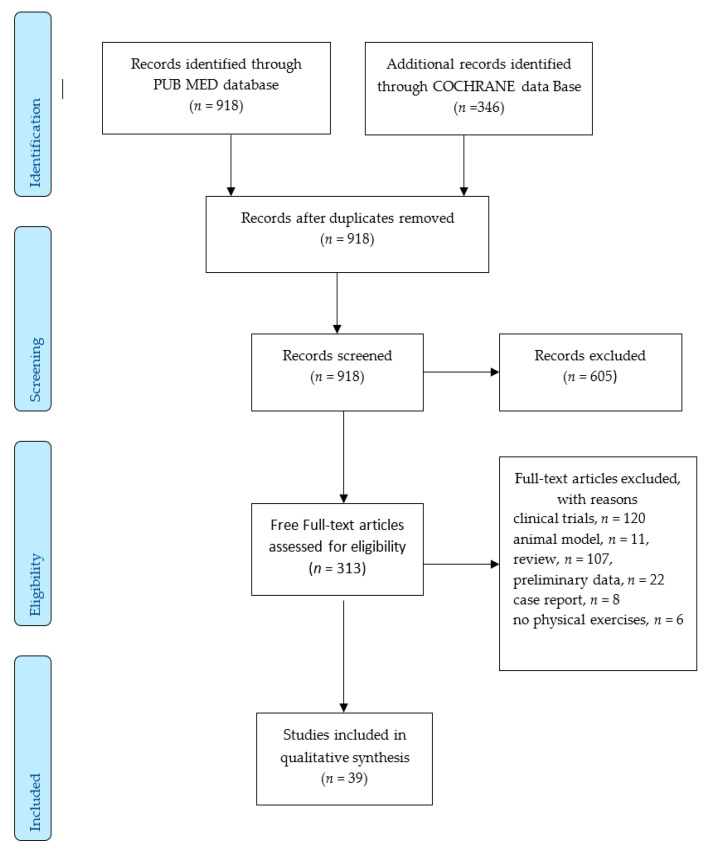
PRISMA diagram.

**Figure 2 jcm-09-03413-f002:**
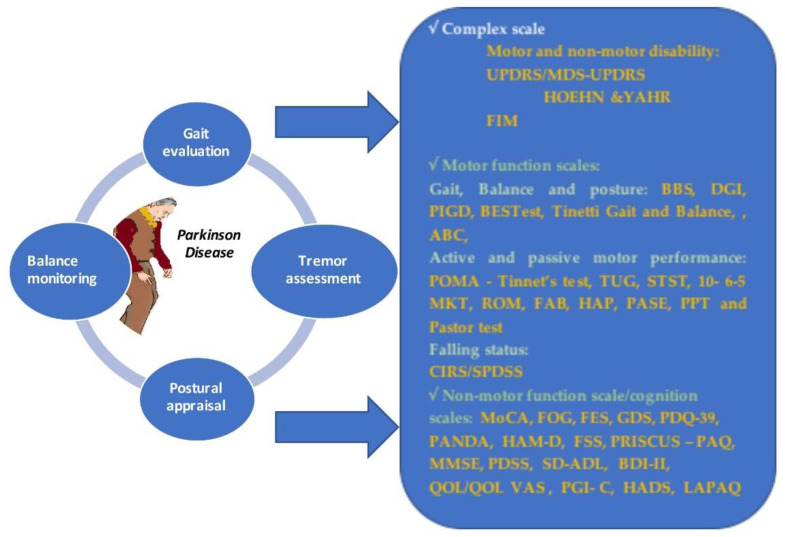
Motor and non-motor instruments used in PD. UPDRS: Unified Parkinson’s Disease Rating Scale; ADL: Activities of Daily Living; PDQ-39: 39-item Parkinson’s Disease Questionnaire; PANDA: Parkinson Neuropsychometric Dementia Assessment; HAM-D: Hamilton Depression Rating Scale; PRISCUS-PAQ: PRISCUS-Physical Activity Questionnaire; Parkinson’s Disease Sleep Scale 2; MMSE: Mini Mental Status Examination; BBS: Berg Balance Scale; FIM: Functional Independence Measure; 0/6 MKT: 10/6 Meter Walk Test; HAP: Human Activity Profile; FSS: Fatigue Severity Scale; ROM: Range of Motion; SPPB: Short Physical Performance Battery; HFLEX: hip flexion range of motion; HABD: hip abduction range of motion; FOG: Freeze of Gait questionnaire; TUG protocol: Time Up and Go test; BESTest: Balance Evaluation System Test; MoCA: Montreal Cognitive Assessment; GDS: Geriatric Depression Scale; PPT: Physical Performance Test; CIRS: Cumulative Illness Rating Scale/SPDDS; HADS: Hospital Anxiety and Depression Scale; LAPAQ-LASA: Physical Activity Questionnaire.

**Table 1 jcm-09-03413-t001:** Summary of the main physical interventions.

Reference	Motor Instruments	Motor Outcomes	Non-Motor Instruments	Non-Motor Outcomes	Physical Exercises	Frequency
Ehlen F., et al. (2018) [57]	UPDRS III HOEHN and YAHR	rehabilitation of motor functions	PANDA HAM-D PDQ-39 PRISCUS-PAQ	improving quality of life	ADL (household, gardening, cycling, walking outside, tidying, gymnastics, reading, watching, sewing)	1 week 2 h twice/week
Frazzitta G., et al. (2015) [61]	UPDRS III HOEHN and YAHR (2–3)	=improving the range of motion improving gait and balance	PDSS—PD MMSE HAM-D	enhancing sleep quality	intensive rehabilitation protocol (occupational therapy, stretching exercises, relaxation, gait training, balance training)	4 weeks 3 sessions/day 5 days/week
Hu Y., et.al (2020) [69]	UPDRS III 10-MKT TUG BBS	improving balance - decreasing neuroinflammation	-	-	aerobic exercises (tai chi, multimodal exercises)	12 weeks 60 min/session twice/week
Kaseda Y., et al. (2017) [70]	UPDRS III-IV HOEHN and YAHR BBS FIM	delaying progressive motor disorders on PD reconverting IV stage in III stage UPDRS after daily exercise improving BBS, FIM	MMSE	increasing quality of life	ADL rehabilitation training (10 m/6 m normal walking)	4 weeks 120 min/session 6–7 days/week
Lana R de C., et al. (2016) [71]	UPDRS III HOEHN and YAHR HAP FSS	rehabilitation motor functions especially on lower limb (bradykinesia) ability to perform daily activities	-	-	ADL	Non available
Medijainen K., et al. (2015) [64]	UPDRS III HOEHN and YAHR ROM SPPB HFLEX HABD	increasing ROM, FOG improving gait speed improving sitting and standing transitions	MMSE FOG	-	gait training—20 m long, 3.5 wide postural exercises stretching exercises	10 weeks 1 h/session 16 session Twice/week
Medijainen K., Pääsuke M., et al. (2019) [44]	UPDRS III HOEHN and YAHR	improving gait, especially speed of lower limbs	MMSE ADL	-	gait self-speed training—12 m TUG test (stand-up, walking 3 m, turn, walking back and then sit down)	1 day
Nascimento CM., et al. (2011) [67]	UPDRS III HOEHN and YAHR	decreasing homocysteine level and improvement of motor functions in PD patients	MMSE ADL	-	Aerobic regularly exercises	6 months 60 min/session 3 sessions/week
Rodrigues-de-Paula F., et al. (2006) [50]	UPDRS III HOEHN and YAHR 10/6 MKT HAP BESTest	rehabilitation of motor functions	MoCA GDS FOG	-	ADL (speech therapy, occupational therapy)	Non available
Stożek J., et al. (2016) [53]	UPDRS III HOEHN and YAHR PPT- Pastor Test (Shoulder Tug)	improving the range of motion improving gait and balance	-	-	gait training stretching therapy breathing exercises balance and gait training dance exercises functional training for head (speech therapy and face exercises), trunk and lower and upper limbs (postural reeducation)	4 weeks I level 120 min/session Twice sessions/day for 2 weeks, II level 120 min/session 1 session/day 3 sessions/weeks for 2 weeks
Trăistaru R., et al. (2016) [52]	GCS FIM Tinetti Gait and Balance Instrument	improving the range of motion	-	improving quality of life	joints mobilization decrease rigidity breathing exercises dynamic balance optimization postural exercises occupational therapy	3 weeks 45 min/session 5 days/week
van Nimwegen M., et al. (2011) [72]	UPDRS III HOEHN and YAHR TUG 6-MKT test CIRS	improving the range of motion improving gait and balance	MMSE FSS HADS FOG LAPAQ–LASA	decreasing fear of falling, depression and anxiety	ADL (walking, cycling, household activities and 2 sport exercises)	2 weeks 111 min/day 1 session/daily
Wang CM., et al. (2018) [73]	UPDRS III HOEHN and YAHR	improving the communication line from swallowing and respiration	MMSE	-	home based oro-lingual exercises training (non-invasive method) 1 cycle (OLE program) 3 times: 2 exercises by swallowing saliva 2 exercises of tongue protruding out 2 exercises of tongue rolling back	12 weeks 3 sessions/day 5 days/week
Wei H., et al. (2017) [74]	UPDRS	reducing the complications of dysphagia recovering swallowing functions	-	increasing quality of life	swallowing management protocol (oral cavity muscles training, tongue training, inhalation exercises)	6 months 10 exercises/session 2 sessions/days

UPDRS: Unified Parkinson’s Disease Rating Scale; ADL: Activities of Daily Living; PDQ-39: 39-item Parkinson’s Disease Questionnaire; PANDA: Parkinson Neuropsychometric Dementia Assessment; HAM-D: Hamilton Depression Rating Scale; PRISCUS-PAQ: PRISCUS-Physical Activity Questionnaire; Parkinson’s Disease Sleep Scale 2; MMSE: Mini Mental Status Examination; BBS: Berg Balance Scale; FIM: Functional Independence Measure; 0/6 MKT: 10,/6 Meter Walk Test; HAP: Human Activity Profile; FSS: Fatigue Severity Scale; ROM: Range of Motion; SPPB: Short Physical Performance Battery; HFLEX: hip flexion range of motion; HABD: hip abduction range of motion; FOG: Freeze of Gait questionnaire; TUG protocol: Time Up and Go test; BESTest: Balance Evaluation System Test; MoCA: Montreal Cognitive Assessment; GDS: Geriatric Depression Scale; PPT: Physical Performance Test; CIRS: Cumulative Illness Rating Scale/SPDDS; HADS: Hospital Anxiety and Depression Scale; LAPAQ-LASA: Physical Activity Questionnaire.

**Table 2 jcm-09-03413-t002:** Available Motor and Non-Motor Instruments.

Tools	Motor Instruments	Motor Outcomes	Non-Motor Instruments	Non-Motor Outcomes	Physical Exercises—Type	Frequency and Duration	References
Accelerometer (Vitaport 3)	UPDRS III ABC BESTest	Analyzing variability gait, gait speed, gait cycles, instability postural	MMSE (˃24)	-	normal walking training around oval inner track by 42 m	1 day 10 min/session 1 session/day	Warlop T., et al. (2016) [75]
A-TPAD (Active Tethered Assistive Pelvic Device) - treadmill Bertec	UPDRS HOEHN and YAHR TUG 5-MKT	- Improving postural disturbances - Decreasing risk of falling - Intervention in balance reactive	-	-	walking training with different speeds on treadmill	30 min/session	Martelli D., et al. (2017) [76]
Biodex Isokinetic—Trampoline	UPDRS HOEHN and YAHR	Rebound therapy exercises improved proprioception, force of contraction for lower extremity muscle and range of movement	PDQ-39	Enhancing the quality of life	- rebound therapy exercises - weight-supported exercises	8 weeks 20–45 min/session 3 sessions/week	Daneshvar P., et al. (2019) [55]
Cycle ergometer - high-intensity cycling - low–intensity cycling with Auditory (metronome) and visual cues—VR—virtual reality	UPDRS HOEHN and YAHR PIGD TUG ABC 10-MKT STST TUG POMA	- Improving performance gait and motor function (akinesia) - Resistance exercises improve postural instability - Endurance physical exercises stimulated connections between thalamus and cortical motor areas - Rehabilitation motor functions by increasing of pedaling rate - Warning for sensory cues for initiating freezing of gait	MoCA BDI-II QOL/QOL PDQ-39 MMSE PGI-C ADL	- subdomain of cognition/emotions were improved through high-cadence cycling training - investigation neurocognitive functions and enhancing emotional recognition - improvement in the quality of life through resistance exercises	cycling training: high intensity low intensity endurance exercises 4-auditory sessions 5-visual sessions with pedaling rate stepping in place and walking on grave for 6 m with auditory cues	1 week 40 min/session 3 sessions/week 8 weeks 15–40 min/session 2 or 3 sessions/week 12 weeks 50 min/session 5 or 2 sessions/day 8 weeks 40 min/session 3 sessions/week 1 day 9 min/session Non available	Harper SA., et al. (2019) [62] Chang HC., et al. (2018) [46] Silva-Batista C., et al. (2017) [47] Shah C., et al. (2016) [77] Gallagher R., et al. (2016) [78] Young WR., et al. (2016) [79]
Dynamometer	UPDRS HOEHN and YAHR	Improving pattern gait (step length), balance and postural stability	-	-	strength training	4 weeks 60 min/session	Shen X., et al. (2012) [80]
GAITRite System (pressure sensitive carpet)	UPDRS HOEHN and YAHR	- Analyzing variability gait, - Improving walking speed and length, wide and time steps so enhances the gait - Repetitive step training improved postural, gait skills and spatiotemporal gait characteristics	FOG	-	normal walking 22 m of which 4.88 m on pressure sensitive carpet and 17.12 m for walking self-speed	1 day 2 sessions/day 15 min/session 3 sessions/week	Shin S., et al. (2019) [81] Shen X., et al. (2012) [80]
GE-O System	UPDRS HOEHN and YAHR	Robotic gait training improved kinematic gait parameters GVSs (pelvic tilt, obliquity, rotation, hip flexion, abduction, rotation, knee flexion, ankle dorsi-plantar flexion, foot regression) but and length, velocity and cadence steps	-	-	gait training	4 weeks 45 min/session 20 sessions (twice session/day for 5 days)/week	Galli M., et al. (2016) [82]
KIN-FOG—system which simulates FOG	UPDRS	The accuracy of the instrument is over 90% in anticipation of the episodes of Freezing of Gait	FOG	-	simple walking (SW) 2.64 m walking with turning (WWT)	3 sessions/day 1 session—SW 2 sessions—WWT (back and forth) 2.64 m/session	Soltaninejad S., et al. (2019) [83]
Lokomat–RAGT (Robot-assisted gait training) with VR (Virtual Reality)	UPDRS HOEHN and YAHR 10-MKT ROM FIM	Improving gait (especially speed)	MMSE (˃24)	-enhancing cognitive parameters	walking self–speed	4 weeks 30 min/session 5 sessions/week	Fundarò C., et al. (2019) [59]
Multidirectional perturbation platform	UPDRS III HOEHN and YAHR DGI PIGD FAB ABC	- Correlation antagonist muscles with reactive balance - Investigating balance, tremor, gait and postural instability	FOG	-	multidirectional disturbance protocol in 12 directions for investigate reactive balance	Not available	Lang KC., et al. (2019) [45]
Neurostimulator DBS	UPDRS	Quantifying gait speed, cadence, stride length, gait cycle	-	-	sitting standing 125 m-normal walking (gait self-speed, back and forth)	1 day - 2 min sitting - 2 min standing - walking self-speed 3 sessions/day	Hell F., et al. (2018) [84]
Opal inertial sensors (feet and belt)	UPDRS HOEHN and YAHR	Turning monitoring (turn mean velocity, amplitude, cadence, duration) of PD patients in daily activities inside and outside	-	-	ADL (daily living activities)	1 week 10 h/day 7 consecutive days	Mancini M., et al. (2015) [85]
Pressure sensor in the object (ball)	UPDRS PIGD FAB	- Turning gait require additional time for secondary task during walking - Delaying the time for installation FOG	FOG	-	4 m - normal speed walking - normal and rapidly speed walking with short steps - turning gait in both directions (360°) with secondary task (rubber ball held in the most affected arm)	1 day 12 sessions/day	Dibilio V., et al. (2016) [86]
Smart EquiTest Balance Master (voluntary stepping in each of the four directions)	UPDRS HOEHN and YAHR	Improving pattern gait, stance and postural stability and decrease fall trends	-	-	walking with directions forward, backward, left and right (sideways)	15 min/session (forward, backward, left and right)	Shen X., et al. (2012) [80]
Step Watch 3 Activity Monitor	PASE	Decreasing gait disorders, improving gait balance, resistance, flexibility	-	- Enhancing the quality of life - Expressing fear of falling	ADL	6 months 20 min/session 3 sessions/week	Ellis T., et al. (2013) [87]
Touch tablet	UPDRS HOEHN and YAHR	Improving writing size and fluency Unchanging writing velocity	MMSE	-	writing training	6 weeks 30 min/session 5 sessions/week	Nackaerts E., et al. (2017) [60]
Treadmill—BWSTT/Bertec	UPDRS HOEHN and YAHR 10-MKT ROM FIM	- Weight-bearing exercises improved proprioception, force of contraction for lower extremity muscle and range of motion - Improving gait speed - Improving step length - Recording steps in PD during overground walking. - Improving pattern gait, stance and postural stability - Improving postural disturbances - Decreasing risk of falling - Intervention in balance reactive	PDQ-39 MMSE (˃24)	- Enhancing quality of life - Improving cognitive parameters	- Weight supported exercises - Walking self-speed - Overground walking - walking with increasing speeds	8 weeks 20–45 min/session 3 sessions/week 4 weeks 30 min/session 5 sessions/week 4 weeks 45 min/session 20 sessions (2 session/day for 5 days)/week 1 day 6 min/session 25 min/session for walking with large step 30 min/session	Daneshvar P., et al. (2019) [55] Fundarò C., et al. (2019) [59] Galli M., et al. (2016) [82] Lai B., et al. (2020) [88] Shen X., et al. (2012) [79] Martelli D., et al. [76]
WEB Neuro (Neurocognitive assessment battery)	-	-	MoCA BDI-II QOL/QOL VAS	Investigating neurocognitive functions and enhancing emotional recognition	Endurance exercises	1 week 40 min/session 3 sessions/week	Harper SA., et al. (2019) [62]
Wearable movement sensors (gyroscope sensor) - gradient tree boosting - LSTM	UPDRS III HOEHN and YAHR	- Estimating severity tremor (UPDRS III) - Recording steps in PD during overground walking	-	-	ADL	120 min/session 2 days/week	Hssayeni MD., et al. (2019) [89] Lai B., et al. (2020) [88]
WII FIT balance board (video game system)	UPDRS III HOEHN and YAHR ABC 10-MKT STST	Improving performance gait and balance	-	-	Nintendo Wii and Wii Fit games	6 weeks 20 min/session twice sessions/day	Zalecki T., et al. (2013) [90]

ABC: Activities-specific Balance Confidence; BDI-II: Beck Depression Inventory-II; QOL: Quality of Life; PIGD: PGI-D: Postural Instability and Gait Disability; STST: sit-to-stand test; POMA: Tinnet’s Performance-Oriented Mobility Assessment; FAB: Frontal Assessment Battery; DGI: Dynamic Gait Index.

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
