# Peer review of "Functional Recovery in Parkinson’s Disease: Current State and Future Perspective"

_jcm, 2020, doi:10.3390/jcm9113413_

Round 1

Reviewer 1 Report

Dear Authors, 

this manuscript is well organised and refers to a sound scientific issue. As all of us know the age-related pathologies together with the increasing aging of global population are, nowadays, pivotal issues that we must address. Although the review field is well represented by the papers searched and included I think that it misses one of the most relevant and actual way in which the rehabilitation take place, that is tele-rehabilitation. Tele-rehabilitation and tele-medicine allow patients interact with therapists remotely and could represent the future of rehabilitation in this field also considering that cost-related issue that affect the health system all over the world.

It is not clear for me why this topic has not been included in the research-strings, however I suggest to add it in order to offering in the state of art, aim of the review, this important perspective. Here you can find some references as examples of how tele-rehab is one of the most actual frontiers:

Telerehabilitation in Parkinson's disease: Influence of cognitive status. Dias AE, Limongi JCP, Hsing WT, Barbosa ER.Dement Neuropsychol. 2016 Oct-Dec;10(4):327-332. doi: 10.1590/s1980-5764-2016dn1004012.PMID: 29213477   Telemedicine in Neurological Disorders: Opportunities and Challenges. Chirra M, Marsili L, Wattley L, Sokol LL, Keeling E, Maule S, Sobrero G, Artusi CA, Romagnolo A, Zibetti M, Lopiano L, Espay AJ, Obeidat AZ, Merola A.Telemed J E Health. 2019 Jul;25(7):541-550. doi: 10.1089/tmj.2018.0101. Epub 2018 Aug 23.PMID: 30136898   Effects of an Innovative Telerehabilitation Intervention for People With Parkinson's Disease on Quality of Life, Motor, and Non-motor Abilities. Isernia S, Di Tella S, Pagliari C, Jonsdottir J, Castiglioni C, Gindri P, Salza M, Gramigna C, Palumbo G, Molteni F, Baglio F.Front Neurol. 2020 Aug 13;11:846. doi: 10.3389/fneur.2020.00846. eCollection 2020.PMID: 32903506   I also suggest to not use abbreviations in the title in order to be more reader-oriented without assuming that all the journal's readers are expert of neurodegenerative pathologies. 

Author Response

Thank you for your time to review and comment. We have considered all your comments and found these very useful in improving our manuscript.

Reviewer 2 Report

This is an exhaustive and very well written review of published studies of interventions aimed at improving quality of life in PD patients with a wide range of procedures and protocols. Strengths of this paper include the organization, flow of the manuscript, fine writing, and balanced assessment of the studies. 

Since the authors' intent is to support the use of these interventions, it may be worth adding a paragraph on the challenges to implementation of these or other services in the real world. At present, any of these worthwhile interventions require insurance approval, and clinicians are in a constant battle to certify and advocate for their patients to get them. There are more significant barriers as well, including access, transport for patients to the places where the therapy is given, and their implementation in settings with challenges to language or cultural acceptance. 

Author Response

Thank you for your time to review and comment. We have considered all your comments and found these very useful in improving our manuscript。

Reviewer Point. Since the authors' intent is to support the use of these interventions, it may be worth adding a paragraph on the challenges to implementation of these or other services in the real world. 

Response: Thank you for the observations that really improved our manuscript. We clarified this in the Discussion part by adding a short comment in line 714

Also, today in this new pandemic situation, the importance of tele health network should be reconsidered. This network allows us to have the best expertise of specialists in the field, not only for diagnostic purpose but also for the rehabilitation. On the other hand, today these services are limited in time and are difficult to access due to insurance limitation. In this light, clinicians spent a lot of precious time with guiding patients to access these services from health authorities. Even in this case, it is well known that a significant part of neurological patients, including PD, can not access the rehabilitation places due to significant functional deficits in the advanced stage of diseases. However, for low or middle income countries across the Europe another significant barrier is represented by the financial support of PD patients by local health politics. All the counties across the EU should come together with the expertise, infrastructure and trainings in order to achieve the best personalized strategies for neurorehabilitation, not only in PD but also in other neurological diseases. The tele heath network across the EU countries can offer a best alternative of the conventional local rehabilitation management, but an disadvantage can be a limited access to all categories of patients independent of level of education, instruction or language barrier.        

and in the conclusion part (line 818)

Also, there is an urgent need to build an European tele rehabilitation network that can be easily accessed by PD patients all over the world from the comfort of their homes that should offer a multidisciplinary strategy for complex diseases such as PD.